# Deciphering DNA Methylation in Gestational Diabetes Mellitus: Epigenetic Regulation and Potential Clinical Applications

**DOI:** 10.3390/ijms25179361

**Published:** 2024-08-29

**Authors:** Nan Li, Huan-Yu Liu, Song-Mei Liu

**Affiliations:** 1Department of Clinical Laboratory, Zhongnan Hospital of Wuhan University, 169 Donghu Road, Wuhan 430071, China; 2Department of Obstetrics, Zhongnan Hospital of Wuhan University, 169 Donghu Road, Wuhan 430071, China; 3Hubei Clinical Research Center for Prenatal Diagnosis and Birth Health, 169 Donghu Road, Wuhan 430071, China

**Keywords:** gestational diabetes mellitus, DNA methylation, epigenetics

## Abstract

Gestational diabetes mellitus (GDM) represents a prevalent complication during pregnancy, exerting both short-term and long-term impacts on maternal and offspring health. This review offers a comprehensive outline of DNA methylation modifications observed in various maternal and offspring tissues affected by GDM, emphasizing the intricate interplay between DNA methylation dynamics, gene expression, and the pathogenesis of GDM. Furthermore, it explores the influence of environmental pollutants, maternal nutritional supplementation, and prenatal gut microbiota on GDM development through alterations in DNA methylation profiles. Additionally, this review summarizes recent advancements in DNA methylation-based diagnostics and predictive models in early GDM detection and risk assessment for subsequent type 2 diabetes. These insights contribute significantly to our understanding of the epigenetic mechanisms underlying GDM development, thereby enhancing maternal and fetal health outcomes and advocating further efforts in this field.

## 1. Introduction

Gestational diabetes mellitus (GDM) is a metabolic disorder characterized by hyperglycemia or glucose intolerance that develops for the first time during pregnancy and typically resolves after delivery. The prevalence of GDM varies significantly worldwide due to differing diagnostic criteria. As of 2021, the International Diabetes Federation estimated a global average prevalence of 16.7% [1]. The incidence of GDM is on the rise globally, driven by increasing rates of obesity among women and societal trends toward delayed marriage and childbearing [2]. GDM poses substantial risks to both maternal and fetal health, including increased chances of gestational hypertension, preeclampsia, fetal macrosomia, dystocia, and a heightened risk of congenital malformations and neonatal hypoglycemia [3]. The impact of GDM on the health of offspring is not limited to birth, but can also develop in long-term metabolic disorders, i.e., obesity/overweight [4], hypertension, dysglycemia, insulin resistance (IR), and dyslipidemias later in life [5,6,7]. This was evidenced by a study following 68,000 women with GDM over 16 years, revealing that 16.2% developed type 2 diabetes (T2D) later in life [8]. Elevated maternal blood glucose levels in late pregnancy have been strongly linked to increased T2D risk in offspring, irrespective of whether GDM was diagnosed during pregnancy [9], highlighting intrauterine hyperglycemia as a risk factor for subsequent T2D development. Indeed, a significant proportion of T2D cases stem from prior GDM [10]. This serves as a significant contributor to the ongoing cycle of T2D. Hence, effective management of GDM during pregnancy is crucial in preventing the future onset of T2D.

IR during pregnancy is a physiological phenomenon in which, under the influence of hormonal changes (placental growth hormone, placental trophins, steroids, etc.), the cell’s response to insulin is reduced, while the pancreatic β-cells compensate for increased insulin secretion to maintain normal blood glucose levels [11]. This adaptive regulation helps protect the fetus from excessive insulin stimulation and ensures that both mother and fetus receive adequate energy and nutrients. It has been reported that GDM patients have higher levels of IR than healthy pregnant women [12]. In GDM, the combination of IR and pancreatic β-cell dysfunction causes insulin secretion to fail to meet its own needs, eventually leading to elevated blood glucose levels. Additionally, genetics play a role in GDM susceptibility [13,14], alongside risk factors such as obesity, advanced maternal age, and familial history [15], exacerbating IR and β-cell dysfunction.

For now, global standardized diagnostic criteria for GDM remain elusive due to racial and ethnic disparities and the absence of universally accepted diagnostic markers. Since 2010, the International Association of Diabetic Pregnancy Study Groups’ (IADPSG) recommended criteria (i.e., the oral glucose tolerance test) [16] have increasingly been used as diagnostic criteria for GDM globally; however, there is still ongoing debate about their usage. The oral glucose tolerance test (OGTT) is conducted after fasting more than 8h between 24 and 28 weeks of pregnancy. The participant is diagnosed with GDM if their fasting blood glucose level is 5.1–6.9 mmol/L, ≥10.0 mmol/L after 1 h of glucose consumption, or 8.5–11.0 mmol/L after 2 h of glucose consumption. The OGTT can recognize GDM that is not detected in routine fasting plasma glucose testing. Overt diabetes mellitus is diagnosed when fasting plasma glucose is ≥7.0 mmol/L or 2 h glucose is ≥11.1 mmol/L during pregnancy, referring to those who are detected during pregnancy and meet the diagnostic criteria for diabetes mellitus in nonpregnant populations. However, several nations and organizations continue to use the Carpenter and Coustan (CC) criteria and the National Diabetes Data Group (NDDG) criteria instead of IADPSG (Table 1).

Early detection and intervention in GDM are imperative, with detection rates in early pregnancy reported between 27% and 66%, influenced by study populations, screening methodologies, and diagnostic criteria [17,18]. In 2023, the American Diabetes Association issued guidelines advocating for early screening and management of GDM [19], emphasizing the importance of identifying risk factors, optimizing pre-pregnancy glucose control, increasing awareness of prediabetes, and implementing early interventions to improve perinatal outcomes and mitigate future T2D risk.

## 2. DNA Methylation in GDM

Epigenetics refers to changes in the level of gene expression based on non-genetic sequence alterations, including DNA methylation, histone modifications, chromosomal remodeling, and non-coding RNA regulation [20]. Of these, DNA methylation is crucial for genome integrity maintenance, gene expression regulation, and chromosomal structure organization [21]. The balance of DNA methylation involves various biological elements, including DNA methyltransferases (DNMTs), DNA demethylases, and methylation recognition proteins (Figure 1).

The dynamic DNA methylation underscores its role in maintaining cellular identity and responding to environmental cues throughout development and life [22]. Aberrant DNA methylation has been implicated in several conditions, including cancer [23,24], neurological disorders [25,26], and developmental abnormalities [27,28] (Figure 1). During pregnancy, interactions between the maternal environment and the fetus can induce changes in DNA methylation patterns that are closely linked to the pathophysiology and clinical manifestations of GDM [20,29,30].

Advances in high-throughput sequencing technologies, including nanopore sequencing [31], methylated DNA-binding domain sequencing (MBD-seq) [32], enzymatic methyl sequencing (EM-seq) [33], and single-cell DNA methylation sequencing [34], enhance the resolution and efficiency of DNA methylation detection. Coupled with evolving bioinformatics tools for methylation site identification and integration with gene expression and signaling pathways, these technologies deepen our understanding of epigenetic regulation in GDM.

### 2.1. Global DNA Methylation in GDM

#### 2.1.1. Placenta

The placenta plays a pivotal role in the exchange of substances between mother and fetus, making it an ideal organ for studying maternal and fetal metabolic disorders. Glucose transporter receptors in the placenta regulate glucose absorption and transport [35]. IR significantly contributes to the development of GDM, exacerbated by gestational age to ensure adequate fetal nutrition. The placenta upregulates endocrine levels during pregnancy, further worsening IR and leading to impaired glucose tolerance. Upon delivery of the placenta, maternal blood glucose levels typically return to normal.

Studies have shown that exposure to GDM alters DNA methylation levels in the placenta [36]. A study using luminometric methylation assays (LUMA) in placentas from 50 pregnant women in the United States revealed lower global DNA methylation levels in GDM patients, which correlated negatively with infant body length and head circumference [37]. In contrast, liquid chromatography–mass spectrometry/mass spectrometry (LC-MS/MS) analysis in placentas from ethnically diverse pregnant women in Germany reported higher global DNA methylation levels in those with GDM [38]. These findings suggest varied DNA methylation patterns influenced by measurement methods and population demographics.

#### 2.1.2. Peripheral Blood

A study by Teresa et al. identified 1141 differential methylation probes (DMPs) in peripheral blood from women with GDM at 26 to 28 weeks’ gestation [39]. Wang et al., using MethylTarget sequencing, pinpointed four CpG sites—89438648 (HAPLN3), 68167324 (RDH12), 157130156 (DNAJB6), and 24837915 (NFATC4)—associated with GDM development in peripheral blood during early pregnancy [29]. Conversely, Dias et al. found no significant difference in global DNA methylation levels in peripheral blood from women with GDM [40]. In addition to different race and gestational weeks, detection methods may vary in coverage, sensitivity, and accuracy, which could affect the consistency of results. As such, further investigations into specific CpG sites versus global DNA methylation in GDM pathogenesis are needed.

#### 2.1.3. Cord Blood

Genome-wide DNA methylation in large cohorts could identify changes linked to disease risk. However, tissue-specific cell types possess distinct DNA methylation profiles, potentially influencing analytical outcomes. Addressing this challenge, Chinese researchers developed the “CellDMC” algorithm to identify cell-type-specific DNA methylation changes associated with GDM risk [41]. “CellDMC” with cord blood samples revealed CpG sites significantly linked to GDM, implicating diabetes-related pathways in a cell-type-specific manner [42]. Standard EWAS, adjusting for cell-type ratios, did not find significant associations, suggesting that differentially methylated loci linked to GDM may be specific to certain cell types. These findings highlight the importance of cell-type-specific methylation changes for elucidating GDM pathogenesis fully.

### 2.2. Gene-Specific DNA Methylation in GDM

Gene-specific methylation investigations enable the discovery of novel mechanisms, targets, and diagnostic markers for understanding GDM. This knowledge lays the groundwork for developing more precise strategies for early detection, treatment, and prevention of GDM.

#### 2.2.1. Energy Metabolism-Related Genes

GDM is closely linked to energy metabolism due to the body’s need to support fetal growth during pregnancy. The placenta serves as a crucial source of nutrition and energy for the fetus. Placental dysfunction in GDM can lead to alterations in the expression of genes involved in energy metabolism within placental tissue. One such key regulator is the peroxisome proliferator-activated receptor-gamma coactivator-1α (PPARGC1A), pivotal for energy homeostasis, with its CpG sites showing positive correlations with maternal blood glucose levels in GDM placentas, notably in relation to OGTT 2 h blood glucose [43].

Another significant gene, the melanocortin-4 receptor (MC4R), plays a critical role in energy regulation and nutritional health. In placentas of GDM-affected women, the fetal side exhibits reduced MC4R DNA methylation levels compared to unaffected placentas [44]. Moreover, DNA methylation levels at maternal MC4R CpG sites correlate with glucose levels [44].

The glucose-6-phosphate dehydrogenase enzyme (G6PD) is essential for cellular energy metabolism and managing oxidative stress. Studies have shown higher methylation levels in the G6PD gene promoter region in blood and placental samples from GDM cases compared to controls. This methylation is inversely associated with G6PD mRNA expression in both blood and placental tissues, as well as with maternal glucose levels during fasting and 1h after glucose ingestion [45].

GDM can alter the expression of genes involved in energy metabolism and influence DNA methylation levels in placental tissues. These changes affect the energy supply and metabolic regulation crucial for fetal development. Such alterations may predispose newborns to metabolic disorders like childhood obesity, diabetes, and cardiovascular disease. Therefore, it is essential to monitor and control maternal blood glucose levels to minimize adverse effects on the baby in pregnancies complicated by GDM.

#### 2.2.2. IR-Related Genes

IR is a central mechanism in GDM. Examination of DNA methylation in genes associated with IR in GDM could offer valuable insights into the development of GDM.

The insulin-like growth factor-binding protein (IGFBP) family genes include multiple members such as IGFBP-1, IGFBP-2, IGFBP-3, and IGFBP-4 et al. These proteins bind to insulin-like growth factors (IGFs) and regulate their biological activity and stability. Studies have shown increased methylation in the promoter regions of IGFBP-1, IGFBP-2, and IGFBP-6 in the placentas from patients with GDM compared to controls, although this difference was not observed in peripheral blood [45]. Furthermore, IGFBP-1 levels in the placenta correlated with insulin sensitivity around 26 weeks of pregnancy, and early pregnancy IGFBP-1 levels in blood can predict GDM [46].

Insulin is crucial for maintaining blood glucose homeostasis by regulating glucose uptake, storage, and utilization in key tissues like the liver, adipose tissue, and muscle. IR in these tissues disrupts this balance, leading to elevated blood glucose levels and contributing to metabolic disorders. Understanding insulin’s specific roles and mechanisms in these tissues is crucial for developing strategies to manage and treat IR and its associated conditions.

Hypoxia-inducible factor 3 alpha (HIF3A) influences glucose metabolism and IR [47]. In samples of omental adipose tissue taken at delivery from women with GDM, the HIF3A promoter CpG island exhibited high methylation levels, leading to decreased HIF3A expression strongly associated with GDM (R^2^ = 0.842) [48]. HIF3A hypermethylation may contribute to IR in pregnant women.

In another study, offspring born to women with GDM showed increased DNA methylation levels of leptin gene (LEP) and adiponectin gene (ADIPOQ) in adipose tissue. Higher ADIPOQ methylation correlated with reduced ADIPOQ and resistin (RETN) gene expression [49]. ADIPOQ has previously been linked to lower IR during pregnancy [50].

Insulin receptors in adipose tissue enhance adipocyte sensitivity to insulin. Insulin binding to these receptors stimulates glucose uptake and conversion to triacylglycerol for storage, while inhibiting lipolysis to reduce fat release into the bloodstream. Impaired insulin receptor function can lead to IR, contributing to GDM development. Studies have found higher DNA methylation levels in insulin receptors’ promoter region in subcutaneous and visceral fat tissues of women with GDM compared to controls [51].

Cell-free DNA (cfDNA) is the nucleic acid fragments released into the bloodstream during cell apoptosis or necrosis, carrying tissue-derived signals. Therefore, detecting differentially methylated CpG loci in serum or plasma samples can serve as biomarkers of cell death. The precise etiology of GDM remains inconclusive, though it is widely believed to be closely associated with IR and β-cell apoptosis. The insulin gene (INS) and islet amyloid polypeptide (IAPP) are recognized markers of β-cell death.

Plasma cfDNA obtained at 12 weeks postpartum showed significantly higher levels of INS DNA in women who had previously experienced GDM and later developed T2D [52]. Another study utilized the demethylation index of islet amyloid in cfDNA alongside insulin to assess β-cell death, linking DNA methylation to IR and newborn weight gain in mothers with GDM [53].

In contrast, a study suggested that β-cell death reduced in women with GDM, indicating that β-cell death itself does not lead to GDM [54]. Unlike patients with T2D, those with GDM typically maintain normal β-cell numbers without destruction. This challenges previous perceptions of GDM causality, suggesting that IR, glucose intolerance, and β-cell dysfunction may play more pivotal roles than reduction in β-cell numbers.

#### 2.2.3. Inflammation and Immune-Related Genes

The interaction between immune–inflammatory dysregulation and DNA methylation is pivotal in understanding GDM. Inflammation is implicated in GDM pathogenesis through contributing IR and pancreatic β-cell damage. Inflammation can influence the activity of DNA methylases and demethylases to alter DNA methylation patterns within cells [55]. These changes could impact gene expression related to the inflammatory system, thereby modifying the function of immune cells.

Placental inflammation is recognized as a contributor to the progression of GDM [56]. Studies have documented a higher ratio of M1 macrophages and a lower ratio of M2 macrophages in GDM placentas compared to normal pregnancies [57]. Under high glucose conditions, placental tissue from GDM patients releases more tumor necrosis factor-α (TNF-α) in vitro [58]. Additionally, obesity, a known risk factor for GDM, is associated with inflammatory changes in adipose tissue that contribute to GDM pathogenesis. Higher TNF-α levels have been observed in visceral adipose tissue of women with GDM, accompanied by significant methylation changes in the TNF-α promoter and increased TNF-α protein levels in the blood [59]. Furthermore, although no DNA methylation changes were detected in the promoter of the suppressor of cytokine signaling 3 (SOCS3) gene, elevated SOCS3 mRNA expression in visceral adipose tissue of GDM patients was linked to altered methylation status in the SOCS3 exon 2 region [59,60].

DNA methylation typically occurs at CpG islands in gene promoter regions, but it can also occur at multiple sites across the genome, influencing gene expression and regulation broadly [61]. Notably, methylation changes outside of promoter regions can significantly impact gene function, as seen in developmental transcription factor genes during early embryonic stages [61]. Understanding these methylation patterns across different genomic regions is critical for deciphering their effects on gene regulation.

The upregulator of the cell proliferation (URGCP) gene, which is involved in inflammatory pathways, has been linked to GDM. Methylation levels of URGCP in cord blood were found to correlate negatively with hemoglobin A1c (HbA1c) levels in the second trimester of pregnant women with GDM [62]. Additionally, IL-10 methylation levels were lower in women with GDM during pregnancy compared to those without [63].

These studies underscore how inflammatory responses can induce DNA methylation changes that impact immune cell differentiation and function, thereby influencing the development and progression of GDM.

#### 2.2.4. Imprinted Genes

Imprinted genes show parent-specific expression that depends on whether it is of paternal or maternal origin. A number of studies have shown that these specific genes are involved in fetal development and metabolism [64]. DNA methylation plays a crucial role in regulating the expression of imprinted genes, and the difference in methylation of the alleles of the two parents creates the silencing of one of the parental alleles [65,66].

With cord blood from infants born to mothers with GDM and healthy mothers, significantly lower expression levels of paternal imprinted genes PEG3 and H19, and maternal genes MEST and MEG3, have been found in mononuclear cord blood cells of infants born to GDM mothers [67]. A recent study has shown that altered methylation and expression levels of MEG3 in the liver of GDM offspring lead to impaired glucose tolerance in adulthood [68].

In another study, IGF2 expression was significantly higher in cord blood and placentas with GDM, while H19 expression was significantly lower in cord blood with GDM [69]. Interestingly, this study also found that the expression of IGF2 is significantly higher in normal glucose tolerance with a macrosomia group compared to normal glucose tolerance with a normal birthweight group, both in placentas and umbilical cord blood [69]. In addition, the methylation level at different CpG sites of IGF2 and H19 in umbilical cord blood was also significantly different among the groups [69]. Otherwise, the methylation of the IGF2/H19 is closely related to birth weight and intrauterine hyperglycemia [69]. This study confirms that intrauterine hyperglycemia exposure will result in altered DNA methylation of the imprinted gene IGF2/H19 in cord blood, which in turn affects gene expression. This change may be a potential pathogenetic mechanism for hyperglycemia-induced macrosomia.

Fetal imprinted gene allele scores (fetal paternally-transmitted INS-IGF2 rs10770125 and rs2585, and maternally-transmitted KCNQ1 rs231841 and rs7929804 alleles) were positively associated with fetal birth weight and maternal glucose concentration, but the association between fetal imprinted gene allele score and fetal birth weight was attenuated after correction for maternal glucose concentration [70]. These results suggest that associations between the fetal imprinted gene allele score and size at birth may be mediated through glucose-dependent mechanisms.

Jiang et al. discovered that intrauterine hyperglycemia causes hypermethylation of IGF2 and H19 in the liver of offspring and downregulates the expression of both genes [71]. Similarly, Ding et al. found downregulation of the expression of the imprinted genes IGF2 and H19 in pancreatic islets from the offspring of GDM animal models, accompanied by altered methylation levels of both genes [72]. These discoveries offer a potential molecular mechanism for glucose intolerance and IR in GDM offspring. Table 2 lists specific genes affected by methylation changes in GDM, providing a comprehensive overview of current research in this area.

### 2.3. Genome-Wide DNA Methylation in GDM

Whole-genome bisulfite sequencing (WGBS) could map methylation status with single-base resolution. This capability is crucial for identifying subtle yet potentially biologically significant methylation changes.

#### 2.3.1. Pregnant Women

During pregnancy, the placenta plays a pivotal role in altering maternal insulin sensitivity through its hormone actions. Sir Ronald Fisher introduced the concept of Mendelian randomization, a causal inference method that utilizes genetic variation to infer the impact of biological factors on diseases [83]. Mendelian randomization analysis of placental DNA methylation sites has identified five specific sites—cg01618245, cg12673377, cg24475484, cg08099672, and cg03699074—that may modulate maternal insulin sensitivity [84]. This approach helps exclude environmental factors’ interference, enabling a more accurate determination of the causal relationship between placental DNA methylation and maternal insulin sensitivity.

Increased body mass index (BMI) is a known risk factor for GDM due to elevated body weight and fat, which heightens IR and insulin requirements, thereby increasing GDM risk. Studies indicate that high BMI may alter DNA methylation patterns, with maternal leukocyte DNA methylation associated with BMI [85] and weight gain during pregnancy [86]. Madelon et al. highlighted differing DNA methylation changes between normal-weight and obese pregnant women, linking maternal glucose concentrations during early pregnancy to specific CpG sites (cg03617420 in XKR6 for normal-weight women and cg12081946 in IL17D for overweight or obese women) [87]. These insights provide valuable clues into understanding the interplay between obesity and GDM.

#### 2.3.2. Offspring

The fetus is exposed to a hyperglycemic environment that can stimulate increased insulin secretion in response. Kasuga et al. identified neonatal hypoglycemia correlated with DNA methylation at two CpG sites near the transcription start site of the zinc finger protein 696 (ZNF696) [78]. Additionally, two CpG sites located in the gene body of Rho guanine nucleotide exchange factor 11 (ARHGEF11) were found to be significantly hypomethylated in cord blood from infants born to mothers with GDM who had large babies [79]. Similarly, genome-wide methylation analysis of cord blood from GDM offspring revealed epigenetic effects primarily on genes involved in pathways related to type 1 diabetes, immune function (MHC-related pathways), neuronal development, and fetal growth and development [88]. These findings in humans underscore the concept of fetal metabolic programming through epigenetic modifications.

In a mouse model of GDM, alterations in DNA methylation levels were observed in signaling pathways related to glucolipid metabolism in the offspring’s pancreas. These changes may contribute to disturbances in glucose and lipid metabolism, thereby increasing the offspring’s susceptibility to T2D and obesity in adulthood [89].

Furthermore, during pregnancy, the fetus develops in the amniotic fluid, which contains exfoliated fetal cells. Analysis of amniocytes has shown that GDM exposure alters genome-wide DNA methylation profiles in a gender-specific manner. Differentially methylated loci interact with genes influencing epigenetics and immunity [90]. Figure 2 illustrates these DNA methylation alterations in offspring with GDM and their potential implications.

## 3. DNA Methylation Changes and the Health of Offspring with GDM

The hyperglycemic environment experienced by mothers with GDM can induce alterations in fetal DNA methylation levels. These changes have the potential to disrupt gene expression patterns in embryos and fetuses, leading to long-term health effects in offspring. Moreover, certain epigenetic modifications may be passed from parents to offspring, a phenomenon known as “transgenerational epigenetic inheritance” [91].

### 3.1. Neurodevelopmental Disorder

Research has shown that offspring of mothers with GDM are at higher risk for neurodevelopmental disorders [92]. This risk is linked to changes in DNA methylation levels of the solute carrier family 6, member 4 (SLC6A4) gene observed both in the placenta [73] and in cord blood [80], potentially impacting the development of the baby’s brain. Similarly, exposure to GDM alters PTEN/AKT/mTOR-regulated autophagic signaling pathways in the brains of children, increasing the risk of neonatal hypoxic–ischemic encephalopathy [93].

### 3.2. Cardiac Function

While GDM exposure does not typically cause changes in normal cardiac function in offspring, it affects the recovery of cardiac function following ischemia [94]. A study demonstrated that offspring exposed to GDM exhibit increased global DNA methylation levels in heart muscle cells. In vitro experiments blocking DNA methyltransferase DNMT3A led to increased expression of cardiovascular Sirtuin 1 (SIRT1) protein, suggesting a potential molecular mechanism contributing to heightened sensitivity to ischemic heart defects [94]. Additionally, research conducted by Su et al. found that elevating DNMT3B in heart muscle cells reduced SRY-related high-mobility-group box 11 (SOX11) promoter activity and mRNA levels, highlighting the role of DNA methylation in regulating cardiac function in GDM offspring [95]. It is important to note that these studies were conducted in animal models, and further research is needed to confirm these findings in humans and understand the specific molecular mechanisms involved in GDM’s impact on offspring cardiac function.

### 3.3. Metabolic Disorder

In utero exposure to hyperglycemia increases the risk of metabolic disorders in offspring, a phenomenon increasingly attributed to epigenetic modifications known as metabolic memory [96]. Recent studies revealed significantly increased global methylation levels and higher histone deacetylase-2 (HDAC2) expression in the umbilical vein endothelial cells from GDM [97]. Such findings suggest that fetal exposure to GDM induces enduring epigenetic changes in endothelial cells, contributing to long-lasting metabolic memory and potentially explaining the health issues observed in GDM children later in life.

Critical regulators of metabolism and pancreatic function, such as pancreatic and duodenal homeobox 1 (PDX1) and peroxisome proliferator-activated receptor gamma coactivator 1α (PGC-1α), are linked to diabetes. Wang et al. identified associations between altered expression and methylation status of PGC-1α and PDX1 in the placenta with glucose utilization in fetuses of women with GDM [74]. IR in GDM not only affects maternal health but also exerts long-term effects on fetal and offspring health through alterations in DNA methylation and gene expression, such as cyclin-dependent kinase inhibitor 2A/B (CDKN2A/B) in rat offspring islets [81]. These findings underscore the complex interplay of epigenetic modifications in response to intrauterine hyperglycemic exposure and their implications for metabolic health across generations.

### 3.4. Obesity

Delta-like 1 homologue (DLK1) gene methylation status in the placenta is not only highly correlated with 2 h OGTT glucose levels in GDM patients, but also closely related to fetal birth weight [75]. In a different study, DNA methylation levels of the MEG3 gene on the mother’s side of the placenta were higher in GDM patients. These levels were also linked to higher glucose levels in the mother’s blood (r = 0.603, *p* < 0.001) and higher weights for both the mother and the baby (r = 0.568, *p* < 0.001) [76]. Lipoprotein lipase (LPL) is a vital enzyme involved in lipid metabolism. In GDM offspring, studies have confirmed that alterations in fetal DNAm levels at the LPL gene locus are associated with anthropometric characteristics at 5 years of age. LPL DNA methylation levels were positively correlated with birth weight z score (r = 0.252, *p* = 0.04), mid-childhood weight z score (r = 0.314, *p* = 0.01), and fat mass (r = 0.275, *p* = 0.04), as well as negatively correlated with lean body mass (r = 0.306, *p* = 0.02) [77].

### 3.5. Fertility

In addition to affecting offspring metabolism, GDM exposure has been linked to reduced future reproductive capacity in female offspring [82]. Elevated glucose levels and fetal hyperinsulinemia associated with GDM can reprogram the cocaine- and amphetamine-regulated transcript (CART) promoter in the ovaries of offspring. This reprogramming makes CART expression more sensitive to leptin levels, potentially predisposing GDM offspring to fertility issues later in life [82]. Animal models have shown that in utero exposure to hyperglycemia can induce epigenetic changes in the F2 offspring of GDM mice. However, it was discovered that this epigenetic memory, carried by DNA methylation patterns, can be erased by a secondary wave of methylation reprogramming in F2 primordial germ cells during fetal development [98]. Juan et al. investigated DNA methylation changes in infants whose mothers had GDM during their first year of life, highlighting this period as critical for epigenetic remodeling. They found that DNA methylation changes induced by intrauterine GDM persist throughout the first year of life [99]. These findings suggest that GDM may exert long-term effects on offspring fertility through alterations in DNA methylation status.

While these studies underscore the role of DNA methylation in mediating the impact of GDM on offspring health (Figure 2), further research is needed to elucidate the underlying mechanisms of this association and explore strategies for potential mitigation or prevention. Table 2 lists DNA methylation changes in some specific genes related to offspring health.

## 4. Environmental Factors Impact DNA Methylation in GDM

Environmental factors such as lifestyle, diet, stress, exposure to pollutants, medications, and toxins can impact DNA methylation patterns, thereby influencing susceptibility to and development of GDM. Conversely, DNA methylation can also modulate sensitivity to environmental factors.

### 4.1. Nutritional Supplements

The micronutrients (folic acid, choline, betaine, and certain B vitamins) are integral to one-carbon metabolism and thus influence DNA methylation processes. Higher levels of maternal erythrocyte folate and vitamin B12 early in pregnancy have been significantly associated with GDM risk [100]. Ismail et al. found a correlation between maternal choline intake and methylation levels of corticotropin-releasing hormone (CRH) in cord blood (r = 0.13, *p* = 0.007) [101]. Additionally, increased maternal betaine intake and serum folate levels were inversely correlated with DNA methylation levels of IGF2 in both cord blood and placenta tissues.

Supplementation with folic acid during pregnancy has shown benefits in animal models, where it ameliorated glucose metabolism disorders in male offspring exposed to lipopolysaccharide [102]. Similarly, fish oil supplementation during pregnancy has been linked to enhanced fetal nervous system development and reduced pregnancy complications. Dilek et al. observed higher DNA methylation percentages in the IGF gene promoter region (specific CpG sites: 1044 and 611) in cord blood from GDM patients who received fish oil supplements compared to those who did not [103].

### 4.2. Pollutant Exposure

Toxins and contaminants can significantly impact DNA methylation by affecting methyltransferase activity, destabilizing DNA methylation patterns, inducing DNA demethylation, causing DNA damage, and influencing the availability of methyl donors [104]. These effects have the potential to contribute to disease development or exacerbate health issues at both individual and population levels.

A cohort study identified a strong association between elevated arsenic levels in household treated water and the risk of GDM. Specifically, an increase of 5 μg/L in arsenic levels raised the risk of GDM by 10% [105]. Ying et al. found that high arsenic exposure alters DNA methylation patterns in the peripheral blood of pregnant women, and identified specific CpG sites whose differential methylation levels can predict GDM risk among pregnant women exposed to arsenic during pregnancy [106].

### 4.3. Gut Microecology

The gut microbiota, an essential component of the body’s environment, has emerged as a significant influencer of DNA methylation status through various pathways. Research indicates that women with GDM exhibit distinct differences in gut microbiota composition and gut metabolites compared to healthy pregnant women [107,108]. Furthermore, alterations in maternal gut flora during GDM may potentially be passed on vertically to their offspring [109], suggesting a transgenerational impact on microbiota-related DNA methylation patterns and GDM risk.

Women with GDM exhibit abnormal gut microbiota compared to normoglycemic pregnant women, with differences persisting for up to 8 months post-delivery [108]. Offspring born to mothers with GDM also show distinct gut microbiota compositions within the first week and continuing up to nine months [110]. Dysregulated gut microbiota, influenced by epigenetic pathways, may contribute to metabolic dysregulation affecting maternal health and offspring outcomes [111].

The interplay between gut microbiology, environmental factors, and DNA methylation could further influence GDM development, highlighting the need for detailed mechanistic studies to understand the pathophysiological processes of GDM and to identify new avenues for prevention and therapy. Figure 3 illustrates how environmental factors mediate GDM development through alterations in DNA methylation.

## 5. Diagnosis and Risk Prediction of GDM Based on DNA Methylation

Detection of DNA methylation markers is straightforward, cost-effective, and tissue-specific, offering high clinical utility for identifying risk factors, early disease detection, and assessing disease progression. In this section, we discuss the potential application prospects of DNA methylation markers in diagnosing and predicting GDM, aiming to improve early diagnosis and prediction accuracy.

### 5.1. Peripheral Blood

Women’s peripheral blood samples were collected at 12 to 16 weeks of gestation and analyzed using genome-wide 450K arrays to determine epigenomic changes in maternal blood DNA. Five CpGs, including COP9 signalosome complex subunit 8 (COPS8), phosphoinositide-3-kinase regulatory subunit 5 (PIK3R5), 3-hydroxyanthranilate 3,4-dioxygenase (HAAO), coiled-coil domain containing 124 (CCDC124), and chromosome 5 open reading frame 34 (C5orf34), showed potential as clinical biomarkers [112].

Combined with machine learning to explore CpG features, this can improve GDM prediction models’ accuracy. The support vector machine model was developed based on selected CpGs (including cg00922748, cg05216211, cg05376185, cg06617468, cg17097119, and cg22385669) within the promoter region, with area under curve (AUC) values of 0.8138 and 0.7576 in the GDM training and test sets, respectively, and 0.6667 in the independent validation set [113].

Another study selected three key differential CpG sites (cg11169102, cg21179618, and cg21620107) using different two public datasets to construct a logistic regression model, which had an AUC of 0.8209 in the test set and 0.8519 in the validation set [114].

Epigenetic markers change as pregnancy progresses. Teresa et al. investigated epigenetic changes at two time points during pregnancy (weeks 24~28 and 36~38) in GDM. They found a stable presence of some epigenetic marks at two time points, and these CpG sites were associated with pathways related to type 1 diabetes, IR, and secretion [115]. Three of these CpG sites (cg01459453, cg15329406, and cg04095097) were able to distinguish GDM cases from controls with high specificity and sensitivity (AUC = 1, *p* = 1.26 × 10^−9^) [115].

Blood samples were collected between 26 and 30 weeks of gestation for DNA methylation testing, which has a potential predictive ability for abnormal glucose metabolism at 4 years postpartum in women with GDM [116]. A recent meta-analysis showed that women with GDM had higher T2DM DNA methylation risk scores than non-GDM [117]. In GDM patients, LINC00917 and TRAPPC9 methylation levels were found to be useful indicators for identifying early glucose metabolism disorders four years after delivery [116]. DNA methylation may therefore be an appropriate tool to identify T2D later in life in women with GDM.

### 5.2. cfDNA

Liquid biopsies, particularly the analysis of cfDNA, have emerged as a promising noninvasive diagnostic approach in oncology. However, no single gene-specific DNA methylation biomarker or panel of DNA methylation biomarkers has been established for early diagnosis of GDM.

Giorgia et al. applied a deconvolution approach to maternal cfDNA methylation data and observed that placenta-specific DNA methylation levels increased early in pregnancy and continued to rise in GDM cases as pregnancy progressed [118]. Interestingly, they found a significant increase in non-placental tissue-derived pancreatic cfDNA fraction only during the first trimester in subjects who later developed GDM. By constructing a classifier for GDM based on cfDNA methylation data, they achieved 80% sensitivity in detecting true GDM positives in early pregnancy [118]. These findings suggest that GDM signs can be detected earlier by analyzing placenta-derived methylation data in maternal cfDNA, offering new opportunities for early diagnosis and intervention.

### 5.3. Epigenetic Clocks

David Sinclair, a prominent aging scientist, published results from a 13-year study in 2023 demonstrating that epigenetic changes are the primary drivers of aging in mammals, and restoring epigenomic integrity can reverse aging signs [119].

DNA methylation patterns, known as “epigenetic clocks”, serve as practical tools for assessing biological age and disease risk [120]. Mothers and offspring with GDM exhibit epigenetic age acceleration [121]. Furthermore, epigenetic age acceleration correlates significantly with fasting insulin in offspring and elevated lipoprotein cholesterol (HDL-C) in mothers [121].

Stephanie et al. recently investigated DNA methylation age in 3–10-year-old offspring exposed to prenatal GDM, finding that accelerated epigenetic aging was associated with cardiometabolic risk factors [122].

Another study observed lower acceleration of extrinsic and intrinsic epigenetic age in newborns exposed to intrauterine GDM compared to those without exposure, with differences more pronounced in female newborns [123]. This suggests varying fetal sensitivity to prenatal environments based on gender.

Lee et al. developed three placental tissue-specific epigenetic clocks for estimating gestational age, unaffected by common pregnancy complications like GDM and preeclampsia [124].

In summary, DNA methylation shows promise in diagnosing and predicting GDM, offering valuable tools for early diagnosis and risk assessment. Figure 4 illustrates DNA methylation’s role in diagnosing and predicting GDM.

## 6. Present Challenges and Perspectives

The regulation of DNA methylation in GDM is still being explored, and its clinical applications require further research. The study initially exposes the potential of DNA methylation in the diagnosis and prediction of GDM. With advances in WGBS, this technology may gain attention in the clinic, but data complexity and accuracy issues need to be addressed.

The methylation profile of fetal DNA in maternal plasma correlates with that in the placenta, facilitating the development of non-invasive fetal-specific biomarkers [125]. In particular, cfDNA analysis shows potential application as a biomarker for the early prediction of pregnancy complications such as preeclampsia, GDM, fetal growth restriction, and macrosomia [126].

Given GDM’s multifactorial nature, single-level research is insufficient. To better understand GDM pathogenesis and make accurate etiological inferences, integrated, multi-level studies are essential. As international public data, analytical platforms, and collaborative efforts expand, research resources will become more plentiful and less costly, thus facilitating comprehensive research on GDM pathogenesis and personalized medicine approaches.

## Figures and Tables

**Figure 1 ijms-25-09361-f001:**
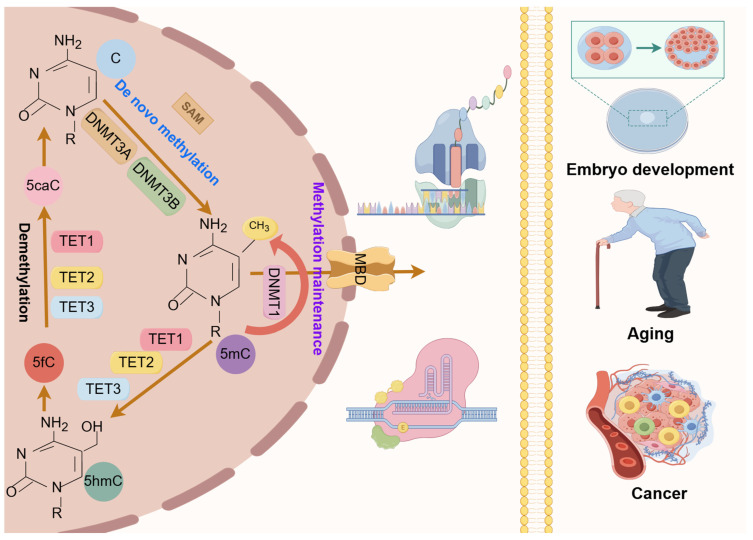
Dynamic DNA methylation and pathophysiology. DNA methylation is a reversible epigenetic mark involving the covalent transfer of a methyl group to the 5-cytosine residue by DNMTs (DNMT1, DNMT3A, DNMT3B). Methylated DNA undergoes dynamic and reversible remodeling through DNA demethylases, namely TET proteins (TET1, TET2, TET3). The dynamic DNA methylation pattern is critical for physiopathological processes such as embryonic development, aging, and cancer. C, cytosine; 5Mc, 5-methylcytosine; 5hmC, 5-hydroxymethylcytosine; 5fC, 5-formylcytosine; 5caC, 5-carboxycytosine; DNMT, DNA methyltransferase; SAM, S-adenosylmethionine; TET, ten-eleven translocation protein; MBD, DNA-binding domain.

**Figure 2 ijms-25-09361-f002:**
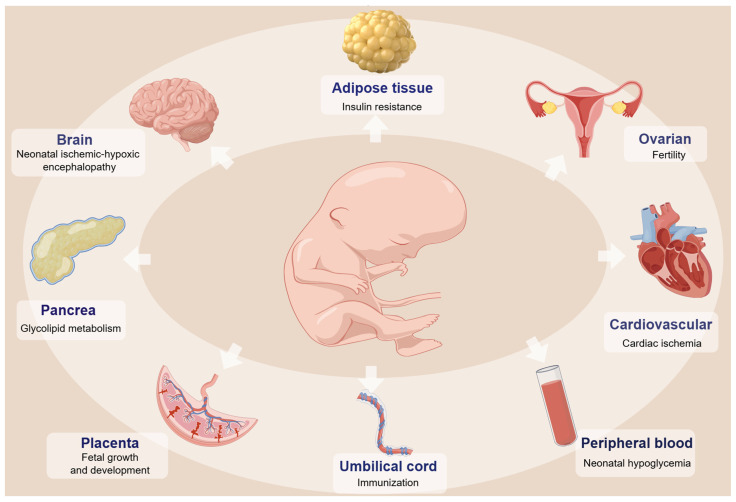
DNA methylation alterations affecting GDM offspring health. Fetuses exposed to GDM present higher susceptibility to various diseases throughout their life. This may be due to the intrauterine high-glucose environment that promotes DNA methylation alterations in the offspring’s organs, affecting their biological functions.

**Figure 3 ijms-25-09361-f003:**
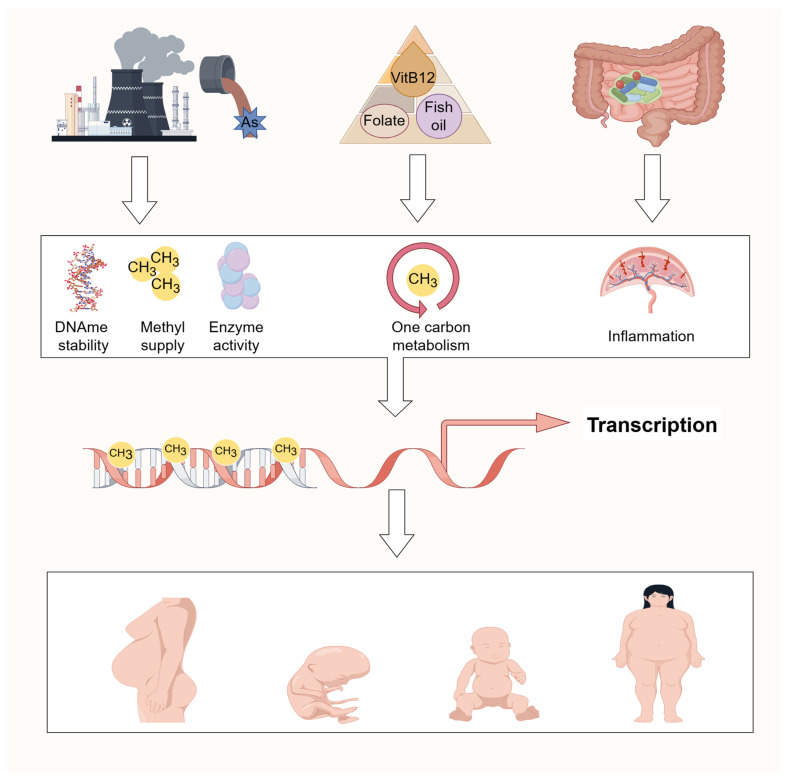
Environmental factors affecting the health of GDM mother and offspring via DNA methylation. In addition to cellular differentiation regulation, DNA methylation could be a genome adaptation in response to environmental stimulations such as chemical exposure, methyl-rich diets, and intestinal microenvironment during gestation. The disturbed DNA methylation could interfere gene transcription and function to affect the health of the GDM mother and offspring.

**Figure 4 ijms-25-09361-f004:**
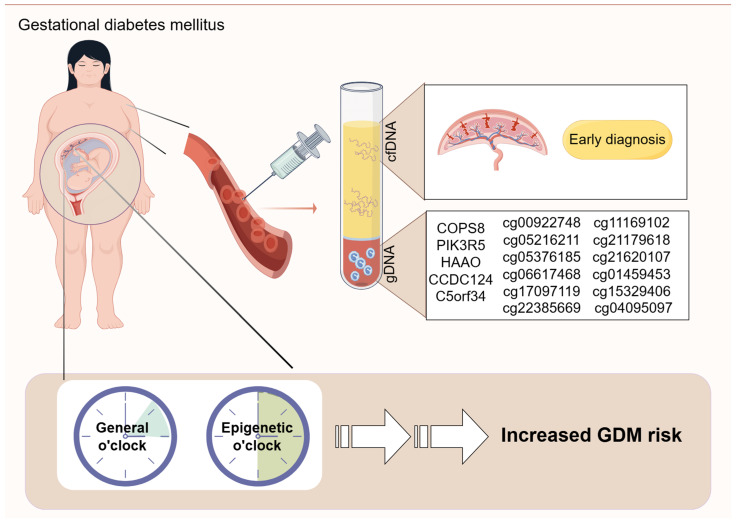
DNA methylation testing in the diagnosis and risk prediction of GDM. DNA methylation testing of genome-wide or specific genes has shown potential for GDM diagnosis or risk prediction. gDNA (genomic DNA) and cfDNA are used as blood-based biomarkers, whereas general o’clock and epigenetic o’clock are placenta-based biomarkers.

**Table 1 ijms-25-09361-t001:** Various diagnostic criteria for GDM.

Diagnostic Criteria	Methods	Glucose Intake (g)	Blood Glucose Level (mmol/L)			
			Fasting	1 h	2 h	3 h
IADPSG	One step	75	5.1~6.9	10.0	8.5~11.0	Null
CC	Two step	100	5.3	10.0	8.6	7.8
NDDG	Two step	100	5.8	10.6	9.2	8.0

IADPSG, International Association of Diabetic Pregnancy Study Group; CC, Carpenter and Coustan; NDDG, National Diabetes Data Group.

**Table 2 ijms-25-09361-t002:** DNA methylation changes in specific genes related to maternal and offspring health.

Samples	Methods	Genes	DNAme	Significance	Reference
Placenta tissue	Pyrosequencing	PPARGC1A	Up	Maternal blood glucose ↑	[43]
	Pyrosequencing	MC4R	Down	Maternal blood glucose ↑	[44]
	MS-PCR	G6PD	Up	Maternal blood glucose ↑	[45]
	MS-PCR	IGFBP1	Up	Maternal IR ↑	[46]
	Pyrosequencing	SLC6A4	Up	Neurodevelopmental disorders	[73]
	BS-seq	PGC-1α	Up	Offspring GMD	[74]
	MethylTarget	DLK1	Up	Maternal blood glucose ↑	[75]
	MethylTarget	MEG3	Up	Maternal blood glucose ↑	[76]
	Pyrosequencing	LPL	Up	Offspring obesity	[77]
Umbilical cord blood	850k BeadChip	URGCP	Down	Maternal blood glucose ↑	[62]
	MassArray	IGF2	Down	Fetal macrosomia	[69]
	MassArray	H19	Up	Fetal macrosomia	[69]
	850k BeadChip	ZNF696	Up	Offspring blood glucose ↑	[78]
	450k BeadChip	ARHGEF11	Down	Fetal macrosomia	[79]
	Pyrosequencing	SLC6A4	Up	Neurodevelopmental disorders	[80]
Peripheral blood	MS-PCR	G6PD	Up	Maternal blood glucose ↑	[45]
	MethyLight	IL-10	Down	Maternal inflammation ↑	[63]
Adipose tissue	BS-seq	HIF3A	Up	Maternal IR ↑	[48]
	Pyrosequencing	Insulin receptor	Up	Maternal IR ↑	[51]
	Pyrosequencing	TNF-α	Down	Maternal inflammation ↑	[59]
	Pyrosequencing	SOCS3	Down	Maternal inflammation ↑	[60]
cfDNA	MS-PCR	INS/IAPP	Up	Maternal IR ↑	[52]
Adipose tissue *	Pyrosequencing	ADIPOQ	Up	Offspring IR ↑	[49,50]
Islet tissue *	MS-PCR	IGF2	Up	Offspring impaired glucose tolerance	[72]
	MS-PCR	H19	Up	Offspring impaired glucose tolerance	[72]
	BS-seq	CDKN2A/B	Down	Offspring IR ↑	[81]
Liver tissue *	Pyrosequencing	MEG3	Up	Offspring impaired glucose tolerance	[68]
	Pyrosequencing	IGF2	Up	Offspring impaired glucose tolerance	[71]
	Pyrosequencing	H19	Up	Offspring impaired glucose tolerance	[71]
Ovarian tissue *	MeDIP qPCR	CART	Down	Offspring ovarian dysfunction	[82]

*: Sample from GDM offspring. ↑: Increased levels of indicators or worsening of symptoms. DNAme: DNA methylation; MS-PCR, methylation-specific PCR; BS-seq, bisulfite sequencing; GMD, glucose metabolism disorder; PPARGC1A/PGC-1α, peroxisome proliferator-activated receptor-gamma coactivator-1α; MC4R, melanocortin 4 receptor; G6PD, glucose-6-phosphate dehydrogenase; IGFBP, insulin growth factor-binding protein 1; HIF3A, hypoxia-inducible factor 3 alpha; ADIPOQ, adiponectin; CDKN2A/B, cyclin-dependent kinase inhibitor 2A/B; SLC6A4, solute carrier family 6, member 4; IGF2, insulin-like growth factor 2; cfDNA, cell-free DNA; INS, insulin; IAPP, islet amyloid polypeptide; ZNF696, zinc finger protein 696; DLK1, maternal delta-like 1 homologue; MEG3, maternally expressed 3; ARHGEF11, Rho guanine nucleotide exchange factor 11; LPL, Lipoprotein lipase; CART, cocaine- and amphetamine-regulated transcript; TNF-α, tumor necrosis factor; SOCS3, suppressor of cytokine signaling 3; URGCP, upregulator of cell proliferation; IL-10, interleukin 10; IR, insulin resistance.

## Data Availability

Not applicable.

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
