# Peer review of "Deciphering DNA Methylation in Gestational Diabetes Mellitus: Epigenetic Regulation and Potential Clinical Applications"

_ijms, 2024, doi:10.3390/ijms25179361_

Round 1

Reviewer 1 Report

Comments and Suggestions for Authors

The manuscript entitled “Deciphering DNA Methylation in Gestational Diabetes Mellitus: Epigenetic Regulation and Potential Clinical Applications” is a narrative review that comprehensively summarizes findings on the role of DNA methylation in gestational diabetes (GDM). The authors made effort to provide a detailed overview of the epigenetic changes in different tissues associated with GDM, affected genes, analytical techniques and factors potentially influencing DNA methylation in pregnancy. The topic of the manuscript is interesting to general readership and it is within the scope of the journal. The manuscript is very well written, while some corrections could further improve the quality of the paper:

- The focus of the Introduction section is on the risk of T2DM (in both mothers and their offspring) associated with GDM. Obesity and other metabolic disorder should at least be mentioned in this section as the most commonly analyzed consequences of GDM pregnancies on offspring health.

- Elaborating on the association of pregnancy with IR and on its role as an adaptive mechanism would be beneficial.

- Diagnostic criteria for GDM are stated on page 3 and in Table 1. However, the authors did not mention oral glucose tolerance test and its diagnostic significance, just the amount of ingested glucose. It is relevant to state that these values refer to OGTT, since this test has certain requirements that need to be fulfilled in order to make the result reliable (like overnight fasting, pregnancy weeks, etc.). The span of glucose levels used for IADPSG criteria should be explained, since it is not a single threshold.

- Lines 79-86: I believe that this section was intended to be a part of Figure legend.

- Section 2.1.2. Potential reasons for discordances between the results obtained by Wang et al. and Dias et al. should be discussed.

- In a section 2.2. more attention should be paid to imprinted loci because of their relevance for pregnancy maintenance, fetal growth and development, the importance of DNA methylation for establishing and maintaining genomic imprinting, as well as because some of the crucial metabolism-related genes are located within imprinted loci and their methylation pattern is proven to be affected by hyperglycemia (like IGF2 and MEG3).  

-  Offspring health is not a suitable headline for a subsection within 2.2. Furthermore, subsections 2.2.5. to 2.2.9. all belong to Offspring health category.

- The headline for 2.3. should be “Genome-wide DNA methylation studies in GDM”, otherwise it would not differ from 2.1.

- Columns in the Abbreviations table are not properly aligned.

Reviewer 2 Report

Comments and Suggestions for Authors

In specialized literature there are numerous articles on this subject. The current study is almost similar to the one published in 2021 in IJMS by Dominik Franciszek DÅ‚uski ( Int. J. Mol. Sci. 2021, 22, 7649. https://doi.org/10.3390/ijms22147649 ) but the topic is presented more at length. The present article has the advantage that it presents the subject in more detail. Also, the bibliography is wider, which is important in the case of a review. In reality, the interest in such articles is low these being more suitable for a book chapter.
Taking into account that mentioned review was considered appropriate for publication, I proposed that present study be published after a major revision.
1. The images should be numerically reduced and redone. Not all of these are necessary.I think they should be less colorful.  Thus: figures 1 and 2 are not necessary and respectively figures 4, 5 and 6 should be graphically redone

2. The bibliography should also include the article IJMS :
https://www.ncbi.nlm.nih.gov/pmc/articles/PMC8303885/

3. It is necessary to edit the text, which includes a series of mistakes.

4.A possible extension of the study would be inappropriate, the review being quite long anyway.

Round 2

Reviewer 2 Report

Comments and Suggestions for Authors

Esteemed Editor and author team,

The study can be published in its current form.

Best regards